# COVID, CITIES and CLIMATE: Historical Precedents and Potential Transitions for the New Economy

**Peter Newman AO** 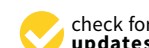

School of Design and Built Environment, Curtin University, Perth 6845, Australia; p.newman@curtin.edu.au

**Abstract:** The 2020 collapse of the global economy due to the Covid-19 pandemic has enabled us to think about long term trends and what the future could hold for our cities and regions, especially due to the climate agenda. The paper sets out the historical precedents for economic transitions after collapses that unleash new technologically based innovation waves. These are shown to be associated with different energy and infrastructure priorities and their transport and resulting urban forms. The new technologies in the past were emerging but mainstreamed as the new economy was built on new investments. The paper suggests that the new economy, for the next 30 years, is likely to be driven by the Paris Agreement and Sustainable Development Goals (SDGs) agendas (summarised as zero carbon–zero poverty) and will have a strong base in a cluster of innovative technologies: renewable energy, electromobility, smart cities, hydrogen-based industry, circular economy technologies, and biophilic urbanism. The first three are well underway, and the other three will need interventions if not cultural changes and may miss being mainstreamed in this recovery but could still play a minor role in the new economy. The resulting urban transformations are likely to build on Covid-19 through "global localism" and could lead to five new features: (1) relocalised centres with distributed infrastructure, (2) tailored innovations in each urban fabric, (3) less car dependence, (4) symbiotic partnerships for funding, and (5) rewritten manuals for urban professionals. This period needs human creativity to play a role in revitalising the human dimension of cities. The next wave following this may be more about regenerative development.

**Keywords:** Covid; waves of innovation; historical precedents; climate; zero carbon; zero poverty

---

"Historically, pandemics have forced humans to break with the past and imagine the world anew. This one is no different. It is a portal, a gateway between one world and the next." Arundhati Roy [1].

## 1. Introduction: What Covid Has Done to Cities

The 2020s began with apocalyptic bushfires across Australia that were unprecedented and harboured the sense that the future had arrived in terms of climate change [2]. However, the Covid-19 pandemic that quickly followed became global and historical in its impact on economies, especially in cities. Not only was it equivalent in impact to the global crash of the 1930s, but it changed our view of what the future could be like. As in other periods of traumatic change, the collapse acts as a catalyst for new markets, and we begin to find hope for long felt visions that could rise out of the ashes of despair [1,3,4]. This paper picks up on the imperative by setting out some historical precedents for how economies have recovered from collapses and created a new world economy where cities were transformed, and a more hopeful future emerged in ways that had been imagined but not truly foreseen.

Cities in 2020 retreated inwards as lockdowns and quarantine became the major focus. However, two transformative changes happened: we telecommunicated in large groups, and we used our local places like never before. There was a huge increase in the use of the internet, computers and mobile phones to create solutions for family connections, conferences, business meetings, shopping

and entertainment. These systems were available before but had not been mainstreamed in their use, especially the ability to have large groups and businesses conduct business-as-usual. Thus, our cities survived, but fossil-fuel-based travel by plane and car reduced dramatically—by 80% in many places [5]. Citizens used local services and places within a short walk much more and could have used a lot more if they were available. The question then is raised whether the economy could recover better if there were greater use of ICT-replaced travel, as well as a range of other localised services that can replace the need for travel. These appear to be two of the factors that could transform our cities.

The paper will pick up this theme and apply it to a range of other available innovations that could be and are likely to be mainstreamed in the next global economy. It will build on the literature for historic economic collapses to show how we can better understand the emergence of innovations; it will then outline what is emerging as the most significant new innovations, especially related to climate mitigation and also transformative economic development. Some will be seen as rapidly mainstreaming and some as much harder to mainstream. Then, it will apply these to how urban environments are likely to be changed in this emerging future, as well as suggesting some potential human dimensions as the basis of more transformative change.

## 2. Historical Precedents for Urban Responses after Economic Collapse

The Russian economist Kondratieff [6] set out the history and future of innovation in a series of economic waves that was used by Schumpeter [7,8] to explain the "creative destruction" of business and the "cluster of technologies" that rose out of economic collapses. Their work was expanded more recently by Freeman and Soete [9] into the theory of long-cycles of economic development. In 2005, Hargroves and Smith [10] built on this and suggested that these waves of innovation could be expressed as in Figure 1 and that the next wave would be related to sustainability innovations. Others like Hall and Preston [11] and Batty [12] suggested that smart city technologies would be the focus of the next wave. In more recent times, it has been the green economy [3,13].

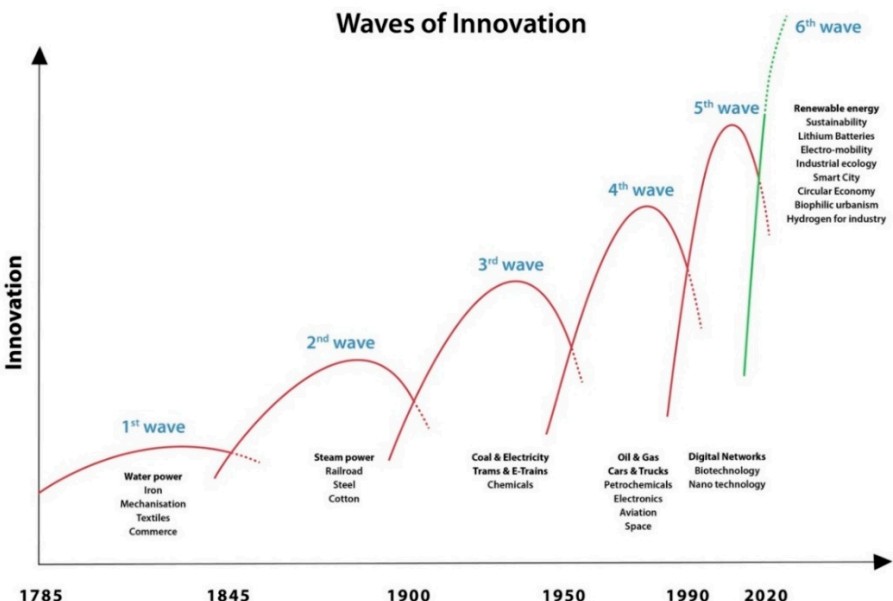

**Figure 1.** Waves of innovation through industrial history and into the future. Source: Adapted from Hargroves and Smith [10].

In Table 1 the ideas developed by Freeman and Soete [9] and Mathews [3] are summarized, showing what has happened after each major economic collapse in western industrial economies over the past 200 years and how five new economic waves grew out of technological innovations that

quickly became mainstreamed with different business models. Added to this table are the associated energy and infrastructure systems that drove each of the economic waves and what that meant in terms of transport and urban form.

**Table 1.** The waves of innovation following economic collapse in the past 250 years and what they meant for business models, energy and infrastructure, transport and urban form, with potential options for the next wave.

| Economic Waves | Technological Innovations Emerging | Business Model | Energy and Infrastructure | Transport and City Form |
|---|---|---|---|---|
| **1. 1780s to 1840s Industrial Revolution** | Water Power Iron Mechanisation Textiles Commerce | Small and cottage industries | Water power and horse power, canals and sailing ship ports; roads for carriages linking cities | Walking cities rapidly densifying from industry |
| **2. 1840s "Hard Times" then Victorian Prosperity** | Steam power Rail Road Steel Cotton | Cottage Industries into Large Capital Firms and Factories | Wood and Steam into Train Systems | Walking Cities into Rail based Linear Urban Development |
| **3. 1890s Great Depression then Belle Epoque** | Electricity Chemicals Internal Combustion Engine | Monopolistic Fordist Firms and Factories | Coal and Electric Tram and Train Systems | Tram- and Train-based Corridors |
| **4. 1930s Great Crash then Keynesian Growth** | Petrochemicals Aviation Electronics Space | Multinationals Modernism | Oil and Freeways | Automobile- based Urban Sprawl |
| **5. 1980s Dot-Com Recession then Knowledge Economy** | Digital Networks Biotechnology Information Technology | Flexible Specialisation and Networked Globalism | Superhighway and ICT Systems | Revival of Urban Centres around Knowledge Economy |
| **6. 2020s Covid Collapse then Green Economy** | Renewable Energy Circular Economy Smart City | Global Localisation. | Renewables with batteries, Electro Mobility especially noncar-based, Smart Cities, Hydrogen for industry, Circular Economy, Biophilic Urbanism | Relocalised Centres, Smart, Distributed Infrastructure, Transit Activated Corridors fed by Micromobility, AVs and Active Transport. |

The sixth wave of innovation is likely to emerge out of the Covid-driven economic collapse. As with the other waves, the innovations began to emerge before the collapse, so Figure 1 and Table 1 list those that could be the new cluster of technologies shaping the new economy. The innovations that appear to be emerging from the ashes of the economy laid waste by the Covid pandemic will be outlined next, some of which may not mainstream easily, as well as how the cluster of innovations are likely to transform our cities of the 21st century through a business model that has been called Global Localisation.

The long-wave theory of how innovations develop has fed into the transition management work by Frantzeskaki et al. [14,15] and Malekpour et al. [16], as well as the sociotechnical transition theory of Geels [17–20]. Both groups are seeking to find how technological innovations can be enabled more

effectively as part of cities, with all their cultural, social and institutional barriers. This research has been very helpful for decision-makers and professionals to see how they need to bring whole system structures into their plans for delivering innovations. They are mostly, however, dealing with a world that is pre-Covid, when there was significant momentum behind the economic system at that time, with lock-in based on investments in infrastructure, buildings, people and manuals of how things should be done [21]. Economic collapses mean there is a discontinuity which is far more significant for opening up economies, cities and culture to rapid change than the incremental understandings in most scenario work [22]. Assets become stranded, and investment shifts to the new options, with potential to last longer. Hence, these barriers may be significantly reduced as the world starts up again because economic collapse may have "cleaned the slate" as Le Corbusier said as he pushed the modernist agenda after the 1930s collapse [23]. However, the barriers may also remain and make change very difficult. This will be examined with the cluster of six technological systems outlined below as emerging options for the recovery of urban economies post-Covid.

The agenda for at least until 2030, but even up to 2100, has been set by two major global agreements from recent years with virtually universal acceptance: the Paris Agreement and the Sustainable Development Goals. However, they are only slowly being adopted. This agenda can be summarised as zero carbon and zero poverty, to be operationally integrated, i.e., to be achieved in symbiosis. The innovations for this agenda are able to attract the cultural and political momentum of something much wanted and needed but not possible until now [24,25]. This agenda has been the subject for much urban transformation literature, particularly stressing the need for new integrated solutions that address the nexus between changes in energy, transport, water, waste and biodiversity and how they shape our cities [26,27].

Zero poverty will be clearly a major agenda as unemployment becomes widespread (e.g., US unemployment rose from 3.5% to 14.7% in the first few months of the pandemic, with much higher levels in the developing world). Underemployment, especially in the emerging nations and cities, will need to be met by significantly more grassroots development and good governance to make it equitable [28]. The opportunity for rapid growth in the new economy is there as Garnaut [29] shows there is a surplus of savings in the world, so the cost of capital is going to be low for many years; hence, substantial long-term loans can drive the next economy. However, the character of such development will be very dependent on factors like the social/equity agenda associated with the investment process and also global factors like the degree of free trade [29].

Zero carbon is also widespread in its support. The demonstrations of the Climate Emergency and Extinction Rebellion are deeply felt, and action has been wanted for some time, with the recent bushfire events highlighting its priority [2]. The time to act more decisively and to make changes may now be here as the economy supporting fossil fuels, with all its support structures, has fallen away and the slate has been largely cleaned for a new economy to emerge. The political and business agenda could now shift to those who see the next few decades as unfolding with a need for more of the innovations that are now clearly on this agenda. However, it may also be a struggle to make such changes happen due to a lack of deliverability. The paper will examine the likely innovations that will lead in this post-Covid new economy, and then, it will assess their deliverability as well as their implications for cities.

## 3. Long-Wave Innovation Theory and Potentials in the New Economy

The next phase of technological innovation will build on what has been emerging, but cities, nations and investors have not been able to break through into mainstreaming the systems that deliver it, just as this has happened in the past. The future that is outlined below can emerge very quickly after a major downturn [1,24] and should not surprise us as scenarios are usually presented without the discontinuities, such as a pandemic, that undermine the economic certainties of the previous era [22,30]. The suggested emerging cluster of innovations for a zero carbon–zero poverty agenda is outlined in Table 1 and below with some of their rationale and early adopters: *renewable energy*, especially

solar PV with batteries; *electromobility,* especially with nonautomobile systems; *smart cities*, especially sensors, apps and ICT focused on localised distribution and efficient demand management; *hydrogen*, especially replacing fossil fuels in heavy industry; technologies associated with *circular economy* and *biophilic urbanism*. These will be adopted through a globally and locally focused economic growth agenda with its urban transformations and, hopefully, with some of the equity concerns of Roy [1] and Sachs [28].

For the global economy to work optimally, it needs to have all the best zero-carbon technologies, as outlined below. However, it also needs to create employment that can enable the zero-poverty agenda. Although in the past, these have been coupled so that economic growth has caused increasing greenhouse emissions, the past decade has begun to show how these can be decoupled [31]. Thus, the technologies outlined below are mostly cost-effective, enabling an agenda where zero-carbon can be achieved along with zero-poverty in new economic development strategies as part of the new economy [32].

Cities will be the focus of such development as history has shown that cities are where economic growth can best be enabled, particularly in the last 200 years [33,34], thus supporting the zero-poverty agenda. It is also where new technologies that are supportive of the zero-carbon agenda are likely to be rapidly adopted, as that is where they are already beginning to be applied. Thus, the zero-poverty and zero-carbon agendas are likely to be rolled out in a synchronistic manner, integrating the economic and environmental agendas. However, it is also possible to imagine how the technologies could be rolled out in a highly inequitable way, and so, the social, cultural and political agendas will always need to ensure that inequitable economic growth in the new economy is avoided [35,36].

### 3.1. Renewable Energy, Especially PV and Batteries

The dramatic growth in renewable energy (solar and wind) globally in the past decade has been due to these technologies quickly becoming the cheapest form of power, as well as being easy to mass-produce and implement in most cities and most economic systems [37,38]. Organisations that predict the future based on previous growth patterns have constantly got this wrong as these are disruptive innovations [32,39], particularly roof-top solar, as it enables very local production and consumption to be integrated [31] and this provides the base for other infrastructure to be localised.

The new patterns of urbanism that are emerging around these systems are already showing why cities will become much more distributed into local areas of infrastructure management, but they will still fit into a citywide or regionwide grid system for equity and balance [40–42]. The rapid growth in solar has now moved into shared solar systems for medium- and high-density housing, enabled by localised solar utilities with batteries and other technologies for enabling sharing such as community-based storage and blockchain-based management [41,43,44]. Industrial estates with shared solar appear to be next, as well as rural and remote systems [41,45].

The next agenda appears to be how to achieve grid stabilisation, and this seems to be heading toward localised, community-scale batteries [41,46–48]. These are becoming available for many other urban functions, including electromobility, which, as shown below, can be part of grid stabilisation. Gas turbines (and diesel backup in small grids) have been seen as necessary for grid stabilisation, but Li-ion batteries are now cost-effective at over 150 MW, showing that they are now cheaper than gas turbines and more effective at providing a rapid peaking function [49]. Electromobility, as shown below, can also help with grid stabilization.

Thus, 100% renewable power systems can now be built cost-effectively. In the US, a report from the Goldman School of Public Policy at UC Berkeley [50] has shown that the US grid can be 90% renewable by 2035 and will be a major generator of employment, with over 600,000 jobs per year being created, replacing 100,000 jobs lost in the old fossil-fuel-based system, thus creating 9 million extra job-years from 2020 to 2035. A report on the Australian grid [51] shows that it can be 75% renewable by 2025, pushed mostly by roof-top solar. Old coal and gas power in the Covid downturn are being avoided as new solar and battery systems have marginal costs of zero; thus, reductions in demand

for fossil fuel-based power must be made. This will only hasten their phase-out. The latest data from the US shows how dramatically this switch is happening due to the sharp decline in demand (Figure 2). This is a real discontinuity not seen by modellers and planners, who have based the future on incremental change.

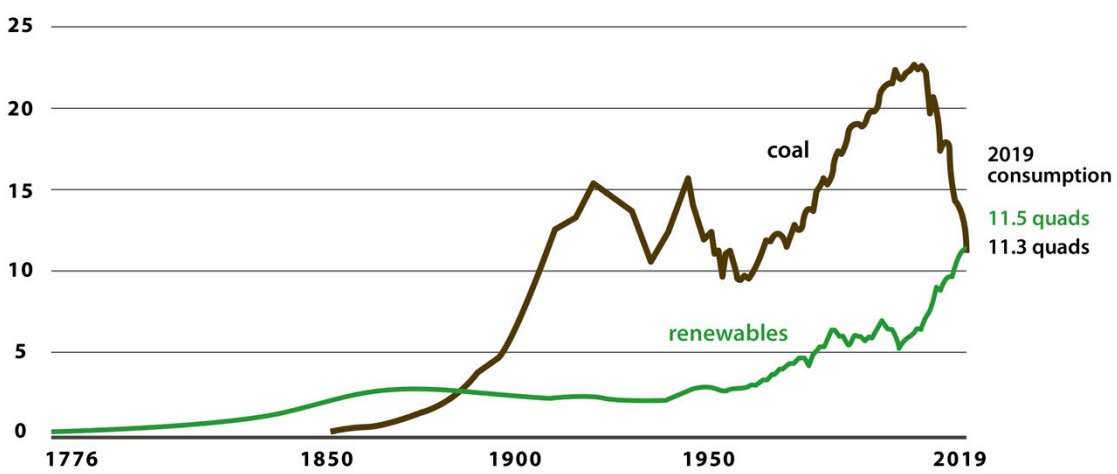

**Figure 2.** The discontinuity already being seen in power consumption in the US. Source: Redrawn from IEA [52].

The same phenomenon of accelerated adoption of renewables has been seen across all 41 nations in the OECD [42], where IEA data showed that in March 2020, compared to the previous year, electricity fell by 3.6% and coal fell by 21.6%, as solar grew 15.6%, wind by 4.0% and gas by a mere 2.7% (and that was nearly all in North America). Such is a classic example of how exponential growth in renewables and batteries can be expected to take over power grids in the near post-Covid future. This is a market-driven process but is helped by the large and growing sector of ethical investing and even the world's largest finance company, Blackrock [53].

The investment in renewables is also enabling new manufacturing centres to emerge with much better local job creation if the products are created from localised renewables [54]. Zero poverty associated with a just transition can also be a goal in areas based around coal, oil and gas, but the cost-effective manufacturing opportunities will rapidly shift to the best sites for renewables [29].

Renewables seem to be well underway and are likely to be a major part of the post-Covid new economy.

### 3.2. Electromobility, Micromobility, Transit, Walkability and Active Transport

Electromobility is also a disruptive innovation that has begun to emerge in recent years [35]. With a growth rate of over 40% per year, predictions for how quickly the internal combustion engine (ICE) will be phased out of new vehicles are now coming down into the 2020s [55]. The reasons electromobility is disruptive are similar to solar and wind: the electric vehicles are rapidly reducing in cost as battery packs have reduced from USD1000/Kwh in 2010 to less than USD200/Kwh in 2020, and the technology is preferred for many other social and environmental reasons, especially cleaning up local air pollution [55–57], with new data showing that Covid viruses are carried by diesel particulates [58–61].

Bloomberg New Energy Finance [62] suggests the capital costs of electric vehicles will be equal to gasoline and diesel vehicles in 2022, and operational costs will be much less. The next electric vehicle

types to roll off assembly lines in large numbers in the early 2020s will be electric buses [63] and electric trucks [64], with some decades before ships and planes.

Concern over the growth in consumption of battery minerals has been seen as a constraint on this rapidly growing Li-ion battery market [65] despite low commodity prices even before the pandemic. Responses to this show that these minerals are needed in very small quantities compared to iron and aluminium and the growth of assured supplies with ethical and transparent mining systems, as well as growing recycling opportunities, suggest these concerns are being addressed [66–69]. However, much more recycling and a much-reduced consumption of energy are necessary agendas for multiple reasons, addressed below in Smart City opportunities.

Disruption is demand-driven, like the smartphone, but those cities adapting to new systems with the cluster of solar-battery electric vehicles (EVs) are also seeing common-good outcomes that they can assist [29]. These public benefits are especially seen in how EVs are assisting electromobility in the new forms of micromobility and new midtier electric transit (often referred to as trackless trams, [70]), which have multiple benefits in overcoming automobile dependence, the double-edged sword of the last big economic wave after the 1930s.

*Micromobility.* Micromobility electric vehicles are all small, local transport technologies that support walking, e.g., electric tuk-tuks, electric bikes, electric scooters and electric skateboards, which are becoming a major part of the EV revolution [71]. The growth of these modes in Chinese cities [72] and places like Delhi (currently the city with the fifth-biggest recharge system in the world, see [73]) is driven by the need to reduce air pollution, but the EVs are also part of the active transport movement that shows substantial health and economic benefits when car dependence is reduced [74]. For example, 46% of car journeys in the US go just 3 miles or less, which can simply be replaced by local micromobility-based journeys; 30% of micromobility riders are doing just that [71].

Electric micromobility can also develop a range of new functions relevant to local centres (a growing focus for the new economy), which could include local delivery of online parcels. The rapid growth in parcel delivery vans has been a major part of growing traffic in the US [75] and this problem can be solved if parcels were delivered locally by micromobility small E-vans that could also be autonomous (as demonstrated in some cities during the Covid lockdown). They can be delivered regionally by trackless trams (see below) along corridors providing parcels to station precincts, where local delivery vehicle hubs can be located.

All forms of electromobility need recharging, and in cities, these can become part of a new recharge hub or battery-storage precinct that is based strategically to support the grid balance needed to ensure universal access and resilience. Such recharge hubs are likely to be driven by power utilities paying for the grid services as well as users' refuelling charges, as are being found in cities moving to electrify their bus-fleets (e.g., Shenzen and Canberra). In Canberra, 60% of their electric bus recharge power costs will be obtained from roof-top solar grid services at bus depots [76]. These recharge services can be available to the multitude of micromobility vehicles in local areas, thus supporting local economies and providing last-mile linkages for electric transit as it services a corridor of economic development (expanded below).

*Transit.* Perhaps the most significant innovation in electromobility in terms of common-good outcomes is the electrification of public transport [77]. The electrification of heavy train and tram systems is a mature technology based on overhead catenaries, but new Li-ion batteries have revolutionised the electrification of buses into electric bus rapid transit BRTs and some, with smart city sensors that guide the system autonomously, into battery-based electric trackless trams [70]. These are now able to fit into cities by enabling the development of new precincts around stations due to their quiet, pollution-free accessibility that is able to replace the equivalent of 6 lanes of traffic. The urban regeneration that is attracted is a high job-creation endeavor that can be used to help pay for the new transit; the resulting station precincts can include recharge hubs for the battery-based transit and micromobility last-mile linkage. They are, therefore, enabling distributed infrastructure and supporting the development of a zero-carbon and zero-poverty city. These innovations will enable the creation of transit activated



corridors [78] along main roads, replacing highly congested car dependence and enabling a series of urban regeneration sites to be developed around station precincts instead of sprawling cities at the urban fringe [79–81]. This has been a major agenda for most cities for the past 20 years but new electromobility is now enabling this agenda and thus is likely to be a big part of future urban economic development strategies, as detailed further below.

*Walkability and Active Transport.* The benefits of active transport in enabling local centres to work without cars and enabling transit systems to work without the need for car-dependent corridors, have certainly rapidly emerged over the past decades [82,83]. Transit was seriously affected during the Covid lockdown, but so was car traffic, and thus, the growth of local walkability and active transport (including the new electric micromobility) has been a global phenomenon, with many cities making this a permanent change [84–86]. At the height of the pandemic in late April, across Scotland, public transport was down by up to 95%, car use was down by 70% and cycling/walking was up by 120% [87].

The co-benefits of active transport are very high, and if local economic development is facilitated, then it is also a part of the zero-poverty agenda. Active transport is hard to facilitate politically due to local car-driver pressure to not lose their spaces for driving and parking. However, the use of these streets by people walking and cycling was so popular during the pandemic that many cities began closing streets to cars and building large numbers of cycleways suitable for all micromobility [85]. This is likely to be a high priority in the recovery period, as in London and Paris.

London has a scenario of increasing active travel by ten times, post-Covid [88]. The mayor and Transport for London said their goals are:

- The "rapid construction" of a strategic cycling network, using temporary materials, with new routes, aimed at reducing crowding on public transport.
- A "complete transformation" of local town centres so that people can walk and cycle where possible, including widening footways on high streets so that people can safely queue outside shops.
- Reducing traffic on residential streets and creating "low-traffic neighbourhoods".

The project was given an immediate GBP2 billion on the basis that "when the nation gets moving again, it does so in a cleaner safer way" [89]. The project also began a special app to help with transport choices, trials on E-scooters (a key part of micromobility), extra charging points for EV vehicles of all kinds, and accelerating new rail projects [90].

In Paris, a commitment has been made to create a 15-min city based on bicycle access by 2024. Local services will be relocalised and infrastructure improved to enable bicycle access in a 15 min-wide precinct. Parking lanes will be given over to bikeways and passing lanes built into shared paths, enabling more urban services to be provided locally [91].

Relocalising a city like this becomes a strong positive outcome of the move to active transport, with its support in micromobility and new electric transit systems, as well as the localised power systems emerging from solar-battery-based power to further the "transformation" of local town centres. It is a sign that a new policy orientation has emerged from this cluster of innovations post-Covid.

Electromobility is rapidly mainstreaming as part of the post-Covid economy.

## 3.3. Smart-Cities-Based Demand Management

Smart cities is an agenda which has rapidly grown in the 21st century. However, it has many controversial aspects if it is used simply for surveillance and control of ideas or as a tool for increased car dependence through traffic congestion management, largely failing due to lock-in or the rebound effect [92,93]. All of the cluster of innovations outlined above—solar, batteries, electromobility in vehicles of all shapes and sizes, and those outlined below as part of the circular economy, water and waste systems—have two key characteristics of relevance to this paper: they are modular, and thus, work best as *localised systems*, and they work even better if *consumption is reduced*. Both can be helped significantly by smart-city-based demand management. Thus, it is possible to imagine why smart

city technologies will become a part of an integrated package of solar-battery–electromobility–smart systems for future cities.

*Localised Systems.* The new smart city technologies include an ability to enable any system to learn and optimise itself through artificial intelligence (AI) or machine learning. Many functions of AI have been envisioned to help the zero-carbon agenda [94], but optimising precinct infrastructure through machine learning is just emerging. As new suburbs, new industrial estates, new office blocks, and new villages are built or refurbished, they can be enabled through sensors to manage their energy, water, waste, transport, and to continuously learn from its users as the data are processed. Thus, the centres become something like a set of neural networks that are constantly improving the ecosystem in which they operate. The roles of recharge hubs and delivery vehicle hubs can all be optimised along with many other new localised services, especially around how energy is managed.

The local precincts set up with such simple technology will be highly efficient and can be optimised to share equitably and enable job creation through enterprise facilitation in local communities [95,96]. The goal is to provide a cost-effective liveability and flourishing economy whilst reducing the consumption of resources, especially fossil fuels. Welfare for people with disabilities or those in aged care can also be improved with these kinds of infrastructure systems. Localised smart systems can be managed to provide different solutions for the different kinds of places across a city and its region, as outlined in Section 4 below.

*Reduced Consumption.* Resource conservation is a critical part of the zero-carbon and zero-poverty agenda [92,93] for better environmental, employment and economic outcomes. Smart technologies can be used to reduce consumption by assisting behaviour change and demand management systems [93]. There is evidence that behaviour can be influenced by direct social interventions by government programs [97,98], but mostly they fail unless they are combinations of social and technological change that provides new knowledge-based systems [99] or they enable infrastructure to work better through enhancing lifestyles in low-carbon urban forms, including changes like relocalisation [81].

These innovations can be called smart-city-based demand management [100]. They enable householders and businesses to understand what they are consuming at any point in time with mobile phone apps and displays in homes and offices, and simple programmable options that build in the optimal efficiencies for the use of energy, water, other materials and transport [101]. These can be apps that use machine language to learn the best options for a local area [102]; they can also be built into new houses and precincts from the start as part of a zero-carbon home or precinct. Zero-carbon transport can use these new kinds of smart system technologies as part of new precincts that are built around new transit systems with walkable environments [103], as shown in transit activated corridors [78].

There is significant economic opportunity for industries to want to be part of this agenda through industrial estates that have become learning environments as they share resources and wastes through industrial ecology [10,45], as well as being able to establish solar- and battery-based embedded electricity networks.

Smart city innovations for achieving better power, buildings and transport are well underway in the newly emerging economy and are likely to be mainstreamed as part of the integrated cluster of innovations.

### 3.4. Hydrogen-Based Industry

The first three innovations are rapidly emerging but the next three are not quite as ready. Big industries are not yet part of the zero-carbon agenda as cement and steel production, as well as most mineral processing, need to use coal and gas for both energy and process chemistry. However, a new option has now begun to emerge and could rapidly move ahead in the new economy. Hydrogen can be electrolysed from water by the use of solar energy and can be stored chemically and mechanically before being used in a variety of ways in industries, as outlined below, and also transport.

### 3.4.1. Transport

The hydrogen fuel cell can enable mobility, but at this stage, it is two to three times the cost of Li-ion battery mobility [104,105], although some possibilities exist for heavy, long-range vehicles like long haul trucks, ships and planes. The distribution system is highly uncertain, whereas power-grids are easily adapted to electromobility based on lightweight batteries. There are small opportunities that should emerge in cities and regional settlements where solar and battery systems produce excess power at nonpeak times, and the extra power can be turned into hydrogen by electrolysing water. This hydrogen, stored locally, could then be used for heavy vehicles using fuel cells for mobility and perhaps grid support as well. These uses are still largely at an R&D stage.

### 3.4.2. Industry

The emerging niche for hydrogen appears to be in industrial activity such as steel and cement production, as well as metal refinement [29]. Hydrogen is a rapidly growing commodity and should be the basis for next economy industrial growth as long as there is space for large solar PV farms or wind farms and a water source to electrolyse for the production of hydrogen used locally to process local minerals and cement production. The competitive advantage of different cities and their regions to be the next hydrogen-based industrial complex is already evident [106,107]. Demand for hydrogen-based refining of metals and cement is rapidly growing post-Covid, as it appears to be the only real option for this large part of the decarbonised future economy and is part of national security considerations [108,109].

The hydrogen part of the next economy appears to be mostly for industrial processing and will depend on rapid R&D, with demonstrations as well as significant changes in large, heavy industries not used to change. These changes suggest a different quality of feasibility and availability for immediate use in the new economy compared to the previous three innovations, although plenty of small niches are likely to find hydrogen useful as part of the cluster of technologies.

### 3.5. Circular Economy

Cities have been innovating for some time to reduce their metabolism, i.e., their resource inputs and their waste outputs [80]. This has been given a new look as the agenda for a *circular economy* [110,111]. The technology for waste disposal in the past has been centralised, large scale and largely linear (not circular), i.e., it has had little emphasis on recycling unless cities have been running out of space. The new systems for the circular economy are, like the other innovations discussed above, much smaller in scale and are able to be used in more localised and distributed situations, even in slum areas [112].

COVID-19 has highlighted the risks associated with an overwhelming reliance on global supply chains and the fragility of these systems. One example is the limited availability of medical supplies to respond to COVID-19 due to an over-reliance on specific countries/producers for global supply. The circular economy movement represents a shift back to production based on localised systems, and extracting value from untapped resources, especially those currently considered waste. This process boosts economic activity and resilience, which are both in dire need for post-COVID communities. If municipal governments were not already facing enough pressure, pre-Covid, to respond to the waste crisis, social isolation practices have driven a significant increase in household waste production [113], meaning the time is now for circular economy innovations to respond to this opportunity.

Despite this being a high priority innovation, there remain significant difficulties with circular economy systems not yet able to cope with new electronic waste, especially batteries, and the ongoing problem of plastics [114]. In many cities, the recycling systems remain significantly less than mainstream, partly due to China banning their imports of waste, but also the sheer thermodynamics of waste recycling [115].

There appear to be deep technical, cultural and economic issues still to be resolved on the cleaner, circular economy despite it being high on political agendas, with opportunities for demonstrating how cities and regions could include circular economy options as part of the cluster of innovations.

*3.6. Biophilic Urbanism, Permaculture and Nature-Based City Planning*

The modernism that helped build cities after the 1930s collapse did not have a strong awareness of the natural environment, or at least how we see it today. It has been 50 years since the first Earth Day showed how the blue planet, viewed from space, was facing new limits that are now described as the Anthropocene's planetary boundaries [116,117]. The biggest part of that agenda is global climate change from fossil fuels and land clearing, which the above technological approaches are meant to address, along with the other SDGs. However, the local natural environment agenda within cities has also been growing and has now coalesced into three areas—biophilic urbanism, permaculture and nature-based city planning.

Biophilia was put on the agenda for cities by E. O. Wilson, who showed that we cannot put ourselves above nature in our cities as we co-evolved and need nature in our daily lives [118]. The response has been a set of technological approaches that has shown how cities can build natural systems into and onto buildings, with green roofs and green walls, which convert water systems from drains to natural watercourses [119–121]. The best examples have been in dense cities like Singapore that have been able to use high rise structures as habitats, like in forests [122,123]. The biophilic cities network is growing across the globe, and it has found that local biophilic features played a very strong role in the Covid lockdown, providing a healthy link to nature [124]. Biophilic urbanism is likely to be a feature of many new urban developments and infrastructure as the cost-effective attractions and local community building potential are high, although it has been slow to be adopted perhaps because it is a break from the modernist way of building cities, with its highly engineered concrete structures controlling natural systems.

Permaculture is similar to biophilic urbanism in its desire to link cities and nature using new kinds of technology and different approaches to making urban ecosystems work better [125]; however, it was essentially developed for suburbs with large spaces around houses, rather than the density where much biophilic urbanism has been applied. The new version of permaculture developed by Holmgren [126] is called RetroSuburbia and is designed to rebuild suburbs around a new vision of the shared economy, food growing and relocalised services. It may emerge with considerably more intensity as middle and outer suburbs begin to be rebuilt, and it can perhaps assist the periurban process of local regenerative farming and food systems. All these processes have been very slow to grow pre-Covid.

Nature-based city planning was propelled into modernist planning by McHarg [127] with his book "Design with Nature" and various applications of this methodology and approach ever since, including ways to use nature as a model for urban development [128] and biomimicry in technological innovation [129]. More recently, the ideas of regenerative design have given a stronger push to these ideas [130–132] though it remains less than a mainstream process in most cities.

These three ecological innovations are not yet mainstream, and despite their low cost, may need a significant structural and cultural change in the design and engineering professions, as well as broader acceptance of what a city can be, before they are taken up significantly in the next economy. They are likely to at least continue to grow in niche demonstrations as part of the cluster of innovations for the new post-Covid economy.

## 4. The Covid Collapse and Multilevel Perspectives on Innovations

The multilevel perspectives (MLP) approach to innovations has grown out of the need to see that innovations are not just technologies but exist in a sociotechnical world [17–20]. There are three levels that distinguish between the innovations: a. those that are not yet well developed and need much more R&D and demonstrations in niche applications (Level 1), b. those with an application across a patchwork of regimes where different interventionist policies are enabling the innovations (Level 2),

and, c. those that are in the mainstream or landscape application where the market takes over (Level 3). Figure 3 below shows where the analysis above would place the six innovations of the new economy.

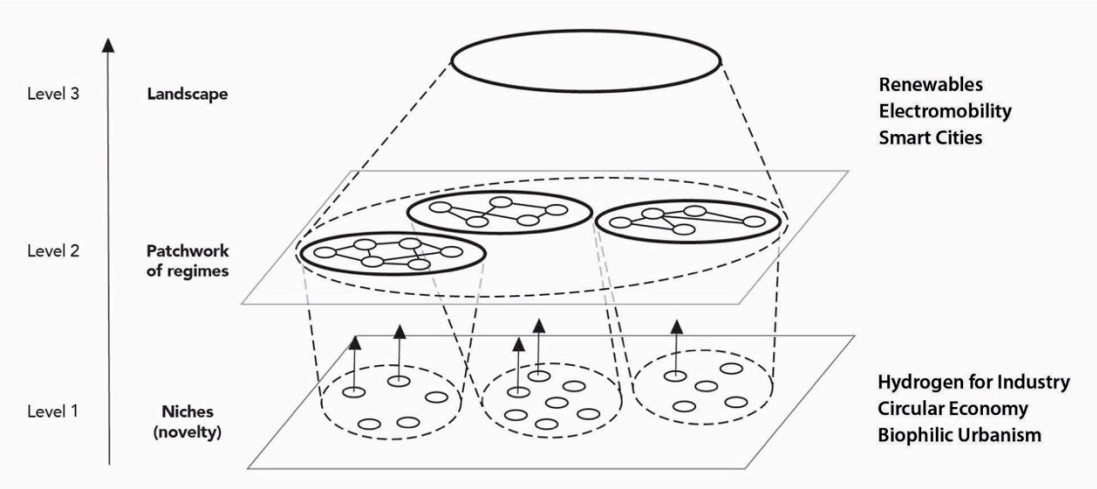

**Figure 3.** Multilevel perspectives on innovations. Source: Redrawn from Geels et al. [17–20].

The first three innovations—renewables, electromobility, and smart cities—are firmly heading towards Level 3, where they will be mainstreamed. These can, therefore, be expected to be the major innovations that shape the post-Covid economy; however, the other three may also play a role. These six innovations (but mostly the first three) will be used below to see how they will shape cities in the next 30 years or so. A key to this is the vertical axis of the MLP figure, which, in the original, was "Increasing structuration of activities in local places". This appears to suggest that the new business model shaping the next economy is likely to be more focused on relocalising the economy, perhaps within a strong global context. This is how the cluster of innovations developed by humankind in the 21st century will be used to create a zero carbon–zero poverty future. Globalised localisation is the business model for the post-Covid economy.

*Globalised Localisation.* The global economy has become a key part of the 20th century, and it is unlikely to lose its appeal as human curiosity for how the rest of the world works will always be there. It is also a world committed to zero poverty by 2030, as well as zero carbon sometime between then and 2050, and this means global partnerships in development. Perhaps the global processes can involve less travel as routine meetings are shifted to computer interactions that have worked well during the Covid lockdown. However, the need for greater focus on localisation in centres that can uptake the new technologies is very clear. The relocalising of places that can enable people to value their place more [133,134] has been a growing social movement for the past 50 years and reached a peak during the Covid lockdown. In this model, businesses will need to work simultaneously on interpreting global trends and creatively addressing local places where they want to invest. The place will be important, and each community will have new tools to make their places better through localised infrastructure management, as well as localised services for a fuller range of activities [134].

The other three innovations—hydrogen in industry, circular economy, and biophilic urbanism—are all needed and have strong groups advocating them and following their progress. From the MLP perspective, they remain on Level 1 where they need a great deal more R&D, demonstrations and a series of cultural and professional barriers to be shifted before they can move from niche applications to regime applications. Hence, we can expect they will continue to grow in this next period as the zero carbon–zero poverty agenda unfolds, driven mostly by the first three innovations.

It is possible the "hydrogen–circular economy–biophilic urbanism" cluster of innovations could be rapidly mainstreamed by concerted action based on strong interventions that drive them into being strong markets. However, it is more likely they will need to wait until the next economic collapse to be mainstreamed, and in the interim, all the R&D, demonstrations and cultural/professional change

could be well underway. This may need the kind of changes being suggested as part of the emerging paradigm of regenerative development.

*Regenerative Development.* Eisenstein [135] suggests there may be downsides post-Covid based on extra-control systems, but he also says that the economic collapse enables us to "embrace holistic paradigms and practices that have been waiting on the margins". Regenerative development is one such holistic paradigm that may be closely linked to the "hydrogen–circular economy–biophilic urbanism" cluster of innovations [32].

Regenerative development is part of a bigger environmental agenda that has been developing for 50 years. It is shown in Figure 4, which sets out how environmental and social impacts have been addressed in three stages and are about to move into the regenerative development phase [136].

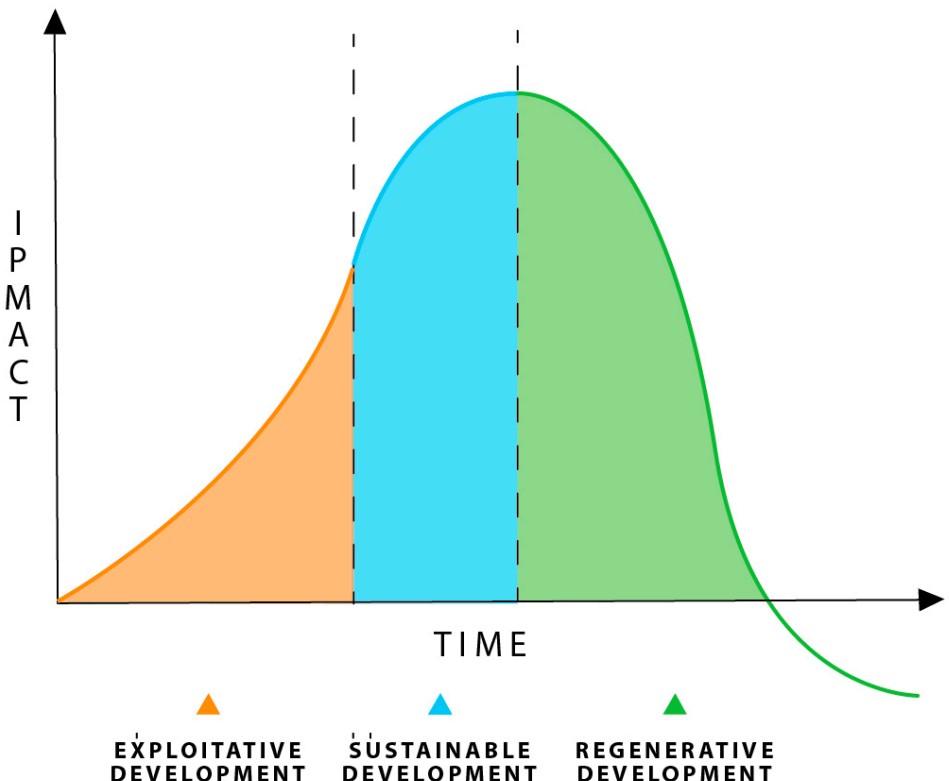

**Figure 4.** The transition in approach to development from exploitive to sustainable to regenerative. Source: Author.

The three stages are

- First, environmental impact began to be addressed through direct regulations and assessment processes from the 1970s, responding to highly exploitive development practices that had developed out of the modernist paradigm.
- Second, reducing impact shifted to the sustainable development paradigm as it was seen that resource issues like energy, water and climate change could not be addressed unless economic and social factors were fully integrated with the environmental agendas; this began to flatten the curve, but it was not transformative enough.
- Finally, we are shifting to regenerative development where the curve turns over and dramatically reduces impact until it is actually able to regenerate the environment [136–139]. In climate terms, this will not just mean zero-carbon but eventually extracting carbon from the atmosphere at a faster rate than it is going in [140].

This is the approach that ultimately makes economic sense while delivering the SDGs and Paris Agreement in a fully mainstreamed way. In this model, not only will power and transport be zero-carbon, but all industrial processes, natural system processes and the whole metabolism of cities and settlements will be like an ecosystem that regenerates itself. There are signs that this regenerative approach may be an emerging paradigm as the need for a transformative outcome becomes more and more obvious [136–140]. Demonstrations are accelerating [141,142], including a new legislated process to make Switzerland a leader in demonstrating the future for aviation fuel by extracting $CO_2$ from the atmosphere and creating a synthetic hydrocarbon-based jet fuel using renewable energy—the whole process is paid for from an airline ticket tax [143].

If businesses, governments and NGOs find themselves compelled by demand from communities to make each place in a city or region achieve regenerative outcomes [130,131], then perhaps the three innovations at Level 1 on the MLP may be rushed from niche demonstrations into landscaped mainstream Level 3. At present, these three seem unlikely to be ready for the present recovery. The regenerative development business and cultural model for economic development may be the only way to enable these three innovations (and others we have yet to imagine) to truly mainstream in a less modernist economic system.

The three innovations that are ready to become mainstream are rapidly rolling out, and with smaller applications of the other three, the whole cluster of six will change our cities in various ways and in different combinations. The next section suggests how this may happen based on an understanding of historical precedents, as with the waves of innovation. The innovations will also be considered to show the kinds of responses by urban professionals to the rapidly emerging new economy.

## 5. The Covid Collapse and Climate Implications for Our Cities

Cities will be the leaders in creating this new economy. There is a long tradition for cities to learn about innovations, modify them and share the results through trade and enterprise development [133]. There are many cities that are already leading in adopting the climate change agenda, with clusters of cities generating global cooperation (100 Res Cities, C40, ICLEI); many will now set their recovery agendas to achieve new goals and outcomes in zero carbon and zero poverty that will establish them as leaders [24,25,144]. This will be the basis of attracting new investment and new people who want to be part of the new economy.

As they adopt the new economic agenda, the technology will flow through into the fabric of the city, just as it has in each other wave of innovation. The fourth wave was based on cars, oil, freeways and housing designed to enable all the new consumption patterns and lifestyle opportunities of the new economy. Cities spread outward and scattered until the digital innovation wave began to assist the reurbanising of old walking city centres and transit fabric [81]. However, the powerful fourth wave remains the source of most of the manuals of modernism that have been used to build our cities.

The next wave is likely to have the following five features, which are summarised in Table 2 and elaborated below.

**Table 2.** Features of the sixth wave in terms of urban responses.

| Features of the Sixth Wave in Terms of Urban Responses |
|---|
| 1.   Relocalised centres with integrated local place infrastructure. |
| 2.   Tailored innovations in each urban fabric. |
| 3.   Less car dependence in most urban fabrics. |
| 4.   Symbiotic partnerships to fund the new urban economy. |
| 5.   Rewriten manuals for urban professionals. |

### 5.1. Relocalised Centres With Integrated Local Place Infrastructure

The three technological systems of renewables (especially solar), electromobility and smart cities make an integrated system of infrastructure. It is a system that works best in local places. Energy can be

created and stored in modular segments to enable homes, industry, education, recreation and transport to be powered and for it to be modulated and managed through smart systems. Each local place system will be joined to each other local place system through a power grid, and together, they can be managed as a system that learns like a set of neural networks.

Planners and other urban professionals have a new focus as they can implement all of these major technological innovations as a system for driving the new economy through *relocalising centres* in most parts of their cities. This is the model being set up in Paris and London through their emphasis on active transport-based urbanism, but it is not a new idea as many cities have been opting for a polycentric urban form in the past few decades [80,81].

There will now be a multiple set of agendas pushing this relocalising process, especially if it has the neural networks of machine learning that can be used for addressing adaptation to climate emergencies or future pandemics. These and other issues depend on mobilising the social capital of a place to build the resilience that will be increasingly needed [145]. Such social capital forms the basis for new job creation activity with local entrepreneurs having the support base they need to create their new businesses [95,96].

Relocalised centres will be very different places, depending on the fabric in which they will be built (as outlined below). The key issue for planners is to ensure that common-good outcomes are set for each localised centre through community engagement and industry partnerships. Common-good outcomes will need to be set in ways that locals can envisage, as well as ensuring that they are dense enough to provide the necessary services [146] and that they are provided with all the required infrastructure, especially transport that is not car-dependent, simply because of space issues [80,81]. The future city urban form is pictured in Figure 5, based on a diagram generated in the 1990s [80] and adopted by many cities but only slowly delivered. The post-Covid drive would now appear to be much more actively pushing this urban planning system.

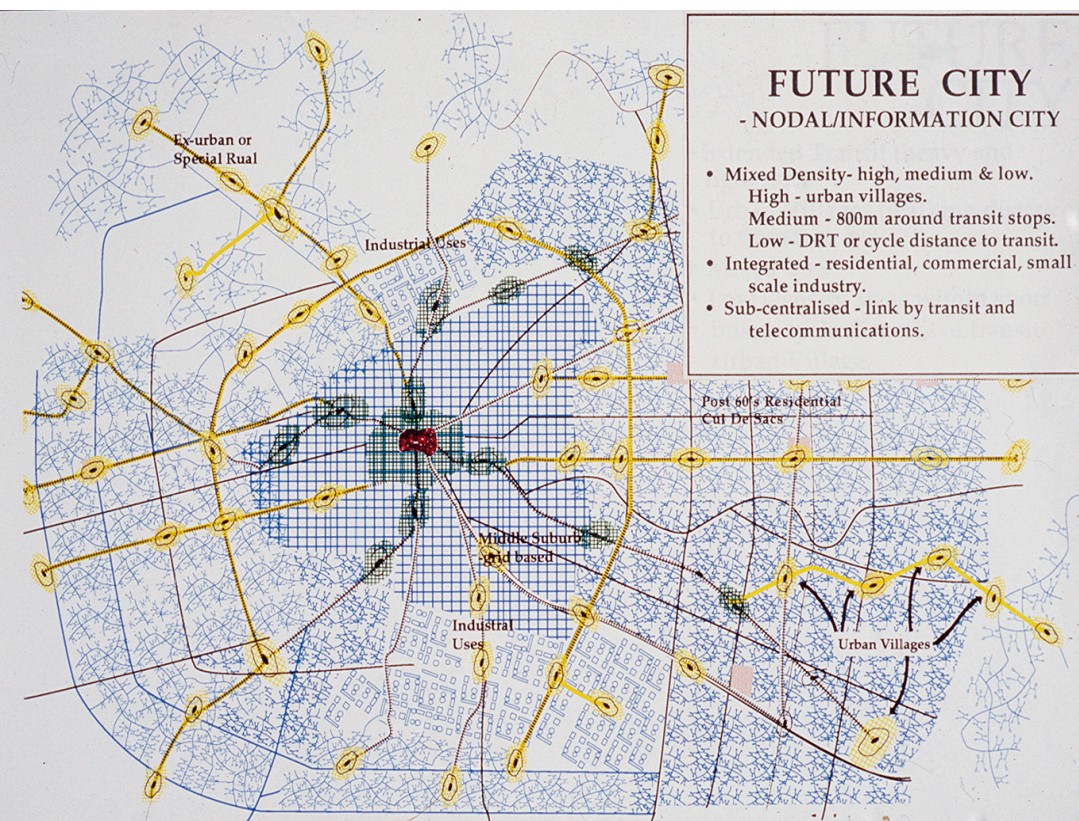

**Figure 5.** Relocalised city of the future based around nodes and linked across the city by quality transit routes. Source: Newman and Kenworthy [80].

## 5.2. Tailored Relocalised Centres in Each Urban Fabric

One of the big problems with the modernism created by Le Corbusier and the Athens Charter in 1933 was that the modernism movement saw "one best way" to create the city of the future. There was one urban system that needed to be modernized, with freeways and housing estates often used as battering rams to clean the slate [147]. Battles resulted throughout the 20th century and have continued up until recently [148]. These responses across the world have shown that walking urban fabric in the historic central area and the transit corridors created by the train and tram eras, as well as the new automobile-based suburbs outside these areas, are significantly different in their urban qualities and, hence, in how they should be managed. The theory of urban fabrics [149] outlines how these three fabrics should be recognised, respected and rejuvenated differently. A further two fabrics were added [32], due to the differences observed in the periurban fringe areas of cities, which merge into rural village fabric [150], as well as remote settlements where mining camps, indigenous settlements and other smaller villages (often temporary recreational sites) are located [2]. Each of these fabrics needs solutions to enable them to create relocalised centres tailored to use their fabric for the best contributions to the new economy.

In Table 3, the core technological approaches that are outlined above are listed down the left column, and the five different types of urban fabric are spread across the top of the table. Inside the boxes are ticks that show the extent to which the core technological approach applies to that fabric. The table illustrates why it is important to tailor the relocalisation innovations differently in different parts of the city and its region, and that together, they make a system that enables the zero carbon–zero poverty agenda to emerge.

**Table 3.** Technological approaches for the new economy and how they relate differently to five types of urban fabric.

| Approaches Outcomes | Walking Fabric | Transit Fabric | Automobile Fabric | Periurban and Rural Village Fabric | Remote Settlement Fabric |
|---|---|---|---|---|---|
| Renewable energy (PV-B) | ✓ | ✓✓ | ✓✓✓ | ✓✓✓✓ | ✓✓✓✓ |
| Electro mobility | ✓ Micromobility | ✓✓✓ Transit and Micromobility | ✓✓✓ Cars | ✓✓ Cars and Farm Vehicles | ✓ Offroad Vehicles |
| Walkability and Active Transport | ✓✓✓ | ✓✓ | ✓ | ✓ | |
| Smart city demand mgt | ✓✓✓ | ✓✓ | ✓✓ | ✓✓ | ✓ |
| Hydrogen for Industry | | | ✓ | ✓✓✓ | ✓✓✓✓ |
| Circular economy | ✓ | ✓✓ | ✓✓✓ | ✓✓✓ | ✓✓ |
| Biophilic urbanism | ✓✓✓ | ✓✓ | ✓ | | |
| Permaculture | | ✓ | ✓✓ | ✓✓✓ | ✓ |

Note: The number of ticks show the extent to which the core technological approach applies to that fabric: none, some, significant, and considerable.

The results in Table 3 are summarized as:

1.  Central walking cities are less able to install solar–PV but are ideal for walkable active transport and micromobility, for smart systems, as well as biophilic urbanism. Circular economy and permaculture do not work well here as they need space.
2.  Transit city corridors are better for solar–PV and batteries, ideal for transit, micromobility and active transport, with some potential circular economy and biophilics, as well as permaculture

possibilities (perhaps in community spaces). Localised activity in centres is already able to be fitted out with these new technologies.

3. The middle and outer suburbs of the automobile era are very good for solar–PV, as demonstrated in Australian cities, where most of the poorer outer suburbs installed PV first [42]. Circular economy and permaculture need more space but are likely to require EV cars and buses due to their car-dependence, along with some new transit activated corridors to help overcome automobile dependence and more relocalised centres (probably in shopping centres) to enable the delivery of new technologies. Industry estates could be eventually enabled with hydrogen-based power (as well as solar), enabling some processing and manufacturing as industrial estates have large roof spaces that are ideal for solar power.

4. Rural villages and periurban areas will need to form new relocalised centres in order to make the most of the benefits of power and transport with integrated solar–PV–batteries–electromobility and with some agricultural vehicles electrified. In these areas, permaculture food production, aligned with local regenerative agriculture and carbon sequestration in soils and trees, presents significant opportunities as well as hydrogen-based industry and circular economy jobs.

5. Remote areas are ideal for microgrids of integrated solar–PV–batteries and electromobility, whether in small indigenous villages or mining camps. A hydrogen-based industry would likely be established adjacent to mining areas, with plenty of space for substantial solar and wind farms to produce the hydrogen.

### 5.3. Less Car Dependence in Most Urban Fabrics

Figure 5 above shows how a polycentric city with relocalised centres linked by transit is likely to emerge, with much-reduced car dependence, based on the need for these centres to be the focus of infrastructure and services in the new economy. The relocalised centres will be made viable or not by the transport priorities chosen to drive the type of urban regeneration sought in each of the five urban fabrics, as each has very different space and time constraints. As Table 3 above shows, there is a need to respect and regenerate each fabric type based on these space and time constraints. For example:

1. The central city needs to be a place for walking as that is its fabric, and the demand for this to continue is obviously growing. Models for this have been created around the world [82]. Pedestrian and cycling infrastructure are the highest priority, following the recent lead of London [84,88] and Paris [91].

2. The transit city needs to be a place for high-quality transit and urban regeneration around its stations. The demand for more of this has created the notion of transit activated corridors along main roads, where new midtier battery–electric transit can move quickly along corridors and slowly through regenerated centres around stations, where micromobility can feed the service and link the surroundings to their local activity centre [78]. Making space on the roads for such high-capacity transit will be the next big issue in transport-prioritising politics.

3. The automobile city needs to be where electric cars are at home, as well as micromobility for linking to new relocalised shopping and services centres, but also to jobs in industrial estates, circular economy/recycling centres, and permaculture food-growing areas. Finding ways to enable local car journeys, but not cross-city journeys, will be a critical disjunction with fourth wave modernist cities. This area can include autonomous electric vehicles, but they are unlikely to be part of walking and transit fabric [151].

4. City-edge fabric or periurban and rural villages need to have a multitude of functions but mostly with a strong local focus; electric micromobility can still work in these areas as it can link to such places for a 10-km radius without being too slow. Electromobility is likely to grow rapidly for such areas, e.g., e-micromobility, EVs and, for agricultural purposes, electric tractors and farm machinery. However, they will still need to grow more compactly in rural villages to enable opportunities for the localised services to be provided, which could include an electric

bus service to help those without electric vehicles [2]. Models such as Witchcliffe Eco-village can easily become the norm as they are cost-effective and on the right side of history in terms of climate-resilient development [152].

5. Remote villages for indigenous and mining functions need to have electric or even hydrogen fuel cell offroad vehicles linked to their solar-based recharge hubs. Many examples of solar villages have been demonstrated in both kinds of settlements, although the pace of change has been slow despite having strong economic and health rationales to replace diesel. Perhaps the new zero-carbon agenda along with the SDGs that work so much better with diesel-free settlements, will be mainstreamed rapidly along with the cities that are presently well ahead of most regions in the decarbonisation agenda [153,154].

In this model, the density of the relocalised "town" centres in each fabric will reduce from the centre to the urban fringe but will need to be compact and walkable, or they will not work further out in rural and remote areas [141,146]. Many mining villages are like medium-density urban villages. Solar and battery recharge hub services will need to be available in all these relocalised centres, as well as delivery hub services, and smart city support services will be needed to enable them to work as zero-carbon places whilst enabling economic activity to flourish.

The developing world will need to approach each of these opportunities as though it is their chance to leapfrog into the future as economic development based around solar–PV–batteries–electromobility–smart city innovations are already challenging traditional development, as they are more appropriately scaled and have multiple benefits [35]. China has led the way in many of these innovations being mainstreamed, and the economic and social benefits are now shown in their cities [155]. The innovations, being small-scale, are also much more relevant to the vast areas of slum developments in the developing world that in modernism were destined only to be cleaned out, thus losing much of their important community structures [112]. These tight structures are ideal for the small-scale energy, water and waste infrastructure opportunities that can be managed locally [156]. Mobile phones have rapidly become part of slums, so in a similar rapid process, developing cities could have better infrastructure in the post-Covid recovery period for both zero carbon and zero poverty. The same pattern across the great megacities of the developing world can pick up on the local community structures that are common to them all and enable regenerative outcomes to be achieved through these new technology systems [157]. Most of all, the next economy will be zero carbon integrated with all the SDGs, including a strong local employment generator that will define the agenda globally and locally.

*5.4. Symbiotic Partnerships to Fund the New Urban Economy*

Relocalised centres and transit activated corridors using new technology systems all need to be funded. Public–private partnerships (PPPs) are the ideal approach for this integrated set of urban redevelopment technologies as long as they have widespread support. The involvement of the private sector is not only because capital is scarce for governments but because the best way to create projects that are viable is around value uplift, where the expertise is best generated by partnerships based on local knowledge and expertise [158,159]. This is found in developing cities, as well as developed cities [160]. Value is created from transit infrastructure, walkability, biophilic features and other urban regeneration features [159–164]. This can now probably be extended to all kinds of new relocalised centres across the city, such as industrial estates, permaculture villages, rural and remote villages, where value can be created by this cluster of technologies, and private sector funding can, therefore, be set up in partnerships to enable the value to be created. The more local value can be added by local communities, the more long-term value will be created [165].

The package of technologies is symbiotic, and so is the need for symbiotic partnerships between all levels of government, the private development industry and local communities. The advantage in creating these symbiotic partnerships is that in the process of using community-based approaches [166], long term commitments are enabled, which form the basis for good government decisions, good

financing arrangements and good community. Community values are fundamental to all viable, flourishing cities [136,167]. With strong community support of zero carbon–zero poverty development, it is likely to be much more feasible. Shared economy demonstrations with symbiotic partnerships are growing in number, and most are based around zero carbon–zero poverty values [165,168].

Cities cannot achieve the zero carbon–zero poverty agenda without substantial redevelopment, especially in their declining middle suburbs where old car-dependent locations may not find the right model for redevelopment [169]. As outlined above, the notion of transit activated corridors could enable this process if set up with symbiotic partnerships to enable urban regeneration for the new economy. This is likely to be the biggest focus of development as the middle suburbs are the area where housing has reached the end of its life. Redevelopment of the "missing middle" with the zero carbon–zero poverty agenda should involve a substantial proportion of affordable, social housing. If the full value of the innovation cluster outlined here (especially the transit systems outlined) are features of such development, then it is likely to receive local and global support that will provide the financial capital, as well as the commitment of social capital.

*5.5. Rewriten Manuals for The New Urban Economy*

The manuals of modernism are alive and well on most urban professionals' desks or at least in their computer models and assessment procedures, based on an idea from the modernist tradition that there is a "one best way" that can be applied across all parts of cities. The need to seek common-good outcomes in planning and urban practices, in general, will still be needed, but there needs to be a new set of manuals created to go with the new economy, especially as they need to apply different approaches to each of the five urban fabrics [170,171]. New context- and outcome-specific zero carbon–zero poverty manuals are needed for planning, transport, energy, water, waste, in fact, every area of the economy developed since the 1940s, with a focus on how these resources can be regulated and prioritised in their infrastructure, density and functions in a far more nuanced way to deliver the cluster of innovations necessary for the new economy [81].

This will not be easy as the sociotechnical transitions literature shows that there are endless barriers already preventing the major innovations being demonstrated for decarbonising the economy [20]. This can be especially seen in the culture of professionals and what they must do to show that their work matters. Rediscovering what matters and delivering projects around that will be an important new task for urban professionals everywhere. Perhaps the triage thinking of Covid professionals can help work out new approaches to assessing projects so that projects do not die due to years of linear, modernist regulatory assessment that is no longer relevant to the new wave of innovations sweeping through our cities and regions [172].

## 6. The Human Dimension: Cultural Urban Renaissance

John Montgomery [4] followed up the theory of waves of innovations that changed economies across the globe in the last 200 years by adding a human dimension—how cultures are part of this change. In particular, they are part of how cities come alive with different cultural expressions and provide the underlying values that enable the city to gain new hope for the future, underlying such activity. Eisenstein [135] suggests that this is now being seen, post-Covid, when he suggests "Covid-19 shows that when humanity is united in common cause, phenomenally rapid change is possible".

What kind of cultural dimension can underly this change towards the cluster of new technologies with their propensity to create relocalised cities and reduced automobile dependence? It will more than likely be local in its orientation, based on a strong sense of local place, and be more urban, not less urban, to enable the human dimension to work in the community. It is also more likely to be deeply embedded in nature-based planning. The fourth wave, based on cars and oil, was a suburban era, and there was a decline in inner and central cities as cities spread out, following suburbanised work, housing and shopping [79–81]. However, the ICT-based fifth wave, which suggested that cities may not even be needed as we could just scatter and telecommute, did not happen; instead, it was the

basis of a new revival of urbanism that was based on the knowledge economy [33]. This required face-to-face meetings for the creative development of projects and soon created the phenomenon of peak car ownership and the creative, millennial generation movement back into more urban locations to live and work [81]. The recent phenomenon of working from home has given rise to the need for more opportunities to have both face-to-face time for creative meetings, and follow-up meetings that can be based on digital face-to-face, thus minimising the need to travel as much. However, the need to continue to have people living closer to work and urban services will be strong foundational pressure on how we build cities. Florida [173] has now suggested that the US reurbanisation of inner areas has faltered as it was not deep enough in its ability to shape the city and rapidly ran out of urban fabric that could be revived. However, the idea of relocalised centres in all five urban fabrics provides the basis for new, more urban development, with a cultural basis in each of the five areas that supports its diversity of forms.

The need for a deeper cultural understanding to achieve the more regenerative features like circular economy and nature-based city planning, as developed above, will need a strong involvement of creative professionals. These perspectives will be driven by the extent to which the formal and informal creative sector is able to help shape urban development; the evidence of recent decades would suggest that this will be with a more local, urban and nature-based flavour and focus. This will be different in different cultures, but without these strong values driving the future, there will not be a fundamental move away from the heavily car-dependent fossil-fuel-based city of the mid to late 20th century, for which much of the creative sector is now looking for something more [82]. Florida [174] and Landry [144] suggest that the creative industry, the most impacted by Covid, is a key contributor to urban economic growth as it enables the city to have a heart and soul as well as a mind, and it builds our sense of place. Now, we sit poised with all the innovation tools that can enable a more robust local, urban and nature-based culture that can help create a more equitable and sustainable city. So, in this model, the creative industry is needed for their more human flavour. Perhaps this city's time has arrived.

## 7. Conclusions

The growth of cities has been the basis of civilization [34], and thus, cities will need to bring together the integrated agenda of zero carbon and zero poverty in a new phase of civility. Cities in the new economy will simultaneously be creating a better global and local environment whilst creating liveability through a better economy. Some commentators have seen a collapse of civilisation [175], and even "civicide" [176], as the only realistic outcome from the pathways that our cities were following until the Covid collapse [136]. Due to the Covid pandemic, things that were ripe for change now have a new opportunity to be mainstreamed. Perhaps the world's cities can create a new kind of civility based on zero carbon and zero poverty using the cluster of technological innovations outlined:

- Distributed renewable energy and batteries, as well as technologies that create distributed energy markets;
- Electromobility and especially the associated new electric transit and micromobility, and the old but tried and true walkability;
- Smart city technology that enables all of these innovations to be integrated, to work better and to create the ecosystems of cities as neural networks that learn and grow, showing us how to make each place in a city or region achieve zero carbon–zero poverty outcomes;
- Circular economy systems applied by construction and recycling businesses in industrial estates and through each local community in the five urban fabrics;
- Biophilic urbanism that brings natural systems on and inside buildings to achieve new urban habitats and green infrastructure, and, with permaculture and nature-based planning, add new ways of growing food and managing natural systems in the various, different fabrics of cities and regions;

- Hydrogen-based industry, replacing the last of the big systems needing fossil fuels: cement, steel and mineral processing.

These changes have been chosen as they appear to be cost-effective, especially the first three, as they have shown their prowess at supporting economies in the time of this Covid pandemic.

The hydrogen innovation is being targeted by many governments as an important opportunity for the industry, in which case it may rapidly move to be cost-effective as it moves up the multilevel perspective transition from niches to regimes of demonstrations [20]. The circular economy and biophilic urbanism/nature-based planning innovations are more deeply cultural and are less developed for professionals to deliver, thus they may need to develop at a slower, deeper pace until the next economic collapse brings them into the mainstream.

Investment will need to have assessment processes that enable rather than drive-away these six core innovations of the sixth wave. Such changes will need new partnership processes that enable all levels of government to work with private finance and business, in close collaboration with communities, whose place-based values need to drive all development and help create local enterprises.

These processes will be a challenge for all urban leaders and professionals who must do more than simply wait for such technologies to solve everything. The need will be for solutions that combine this cluster of technologies with place-based designs: in dense city centres, inner-city corridors, suburban centres with estates for various functions, and periurban, rural and remote villages. The human qualities and natural qualities of each place will be at the heart of how these new technologies can create relocalised places.

Urban professionals will soon see that they need to rapidly change the manuals of modernism that are still so prevalent in their fourth wave engineering designs and statutory regulations, or else they will miss their early chances to be part of the sixth wave. Cities that can quickly focus on how to mainstream their new planning and assessment systems to create new centres of zero carbon–zero poverty urbanism are likely to be the new centres of civilisation, especially if they can work synchronously with the creative side of their cities [144].

**Funding:** This research received no specific external funding but has been part of a long journey of research projects over many decades mostly from Australian Government sources.

**Conflicts of Interest:** The author declares no conflict of interest.

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
