# Peer review of "COVID, CITIES and CLIMATE: Historical Precedents and Potential Transitions for the New Economy"

_urbansci, doi:10.3390/urbansci4030032_

Round 1

Reviewer 1 Report

This paper discusses the opportunities for Cities as they react to the COVID-19 pandemic and its aftermath. The paper suggests the pandemic could be the prompt for a '6th Wave of innovation', based around renewable energy, electromobility, hydrogen economy, biophilic urbanism and a circular economy, which will lead to/be led by a business model focused on globalised-localism and regenerative ecosystems.

The article is perhaps best described as a perspective or viewpoint style article, rather than primary research and will be reviewed as such. It draws on existing ideas and research (including much of the authors' own work) to describe an "expected transition". Inevitably there is subjectivity in any discussion of future transitions. Thus, this review should be seen as an attempt to constructively engage with the author's ideas and how they are presented, rather than a challenge to the intellectual ideas presented. (This should possibly be true of all referee reports!).

I think the article is timely, of course, as it considers the outcomes of the substantial changes we potentially face due to the COVID-19 pandemic. It joins a number of other perspective/viewpoint-style articles coming out that looking at this from the many different research perspectives and adds to these ongoing discussions. It aims for quite a holistic view, which I think is good, and draws together several threads around the possibilities of innovation and technology. I admire the positive transition the author presents - I very much hope the vision they put forward is accurate! However, I would suggest it needs some revision in framing and a refining of the argument it presents before publication.

I have selected minor revisions, because I think there are ways to reframe what is there into a good article, and the topic is timely and interesting. Maintaining the current framing of the article could require more significant revisions.

I highlight 3 related issues that I think there are: (1) it is currently a bit uneven in its discussion of the innovations; (2) it changes between being a prediction, a hypothetical opportunity, and advocating for a vision; (3) it doesn't sufficiently discusses the challenges in the shift that it proposes.

I will address each of these in turn, hopefully in a constructive manner, and then provide an overall critique with suggestions for how the author could address them. There are no doubt many other ways to address them, and the author may choose to reject/push back on some.

Uneveness in Discussion
This is perhaps best highlighted by section 3, where 3.1, 3.2 and 3.3 are very detailed on renewable energy, electromobililty and smart cities (the authors 'wheelhouse' I think), but 3.4, 3.5 and to some extent 3.6 have much less discussion. The circular economy, in particular felt very short and light in details, especially given the later sections on business models and the regenerative economy. Also, to my mind, it is hard to see the link between the pandemic and the circular economy (which is not to say there isn't one). Are there outcomes of the pandemic that are pushing us towards a circular economy? Similarly are there links to the other innovations, e.g. renewable energy or electromobility? I would argue that many current innovations are taking off because they fit neatly into our linear economy.

I agree that the circular economy is an essential transition, but don't see significant evidence much evidence for this in the discussion, compared to renewable energy. The biophilic and permaculture section is better but the transition to these is still not as strongly evidenced as the others. Currently it feels a bit like they have been added as an afterthought (I'm not suggesting they have, this is just how it reads).

Possibly part of the problem is that very different types of things are presented as an equal list. Is the list discussing things at the same level? Renewable energy is a collection of technologies; Smart cities are a particular application of a wide range of technologies or even a urban development paradigm; circular economy is probably an economic paradigm or an economic system; biophilic urbanism could described as an agenda(?).

These sections need rebalancing. However, this uneveness extends to the section in urban fabrics.

This urban fabrics discussion provides the link to the urban and the framework for how these general innovations and technologies can be applied. The regenerative elements feel like a very small part. For example, the only mention of regenerative in this section is:
"These [re-localised centres] can become Regenerative Outcomes if smart city systems are used to learn how this can be done in ways that locals can envisage as well as ensuring they are dense enough to provide the necessary services (Newman, 2016) and they are provided with all the required infrastructure, especially transport". There is no explanation of how they can become regenerative. I think they could easily not be regenerative. And "The same pattern across the great mega cities of the developing world can pick up on the local community structures so common to them all and enable regenerative outcomes to be installed through these new technology systems (UNCRD, 2019)." The link between common 'community structures' (which needs more explanation itself) that appear through these (localised) techonologies and regerenative economies is not clear.

The mentions of regeneration are often urban regeneration - to my mind that is something very different to a regenerative economy.

Similarly, Section 4.5 is about the shift to the new urban economy but doesn't mention circular economies or regenerative economies. It suggests that Public Private Partnerships (PPP’s) are the ideal approach. But it does not explain how they might lead to regenerative economies. Personally, I'm not sure PPP (in the current economy) would provide for regenerative outcomes because private capital requires return on its investment. The article states "Value is created from transit infrastructure, walkability, biophilic features and other urban regeneration features (McIntosh et al, 2017; Matan and Newman, 2016; Cabanek and Newman, 2016)." But how is this value captured in a PPP? The benefits of regenerative outcomes are not always going to be translatable into profit, which drives the private part of the PPP.

Changing focus
As I read I felt the tone of the article changed between being a prediction, a hypothesis and a call for action. I think it could be any of these. The title is 'expected' transitions and it begins very confidently predicting the sixth wave and what it will look like. See, for example, phrases like "The sixth wave of innovation is now likely to emerge out of the Covid-collapse. As with the other waves, the innovations began to emerge before the collapse so it should be possible to see which are likely to be the basis of the new economy". There is a confident 'prediction' that our response will be guided by the SDGs and the Paris declaration. Later in the article it reads more as exploring and explaining the possibilities that technological and innovative shifts provides us. The 'endless barriers' mentioned in section 4.5 suggest this is more of a vision that could be enacted if we can overcome the barriers. The conclusion talks about the "opportunity" we have and perhaps implies it is advocating for this transition.

I think it could be any of these three, but I think the author should be clear throughout what they are trying to present. Personally, I think it is best framed as discussing the opportunities (and challenges) or as a hypothesis that links the various technologies and innovations we have to the potential outcomes (globalised-localism and regenerative economies). In this way, things like the circular economy and even the cultural element discussed towards the end can be highlighted as challenges or things that need to also change. The author may want to advocate for the transition they present as well.

Addressing the challenges
For me, this is the biggest issue. I like the positive view, but I felt it failed to really discuss or address the barriers. Only really in section 4.5 are the "endless barriers already preventing the major innovations being demonstrated for decarbonizing the economy" mentioned  let alone the barriers to a regenereative enconomy. And here they are largely related to the "culture of professionals". The political, economic and governance barriers are enormous. We see already the Australian government talking about a gas-focused recovery (despite the many calls for a renewables' focused one). There are issues with special interests. The localism element might circumvent some of this, but only if there is radical change in governance. Government or privately-owned electricity companies have no interest in letting us generate our own power!

There are links here to my criticisms around the lack if discussion on the circular economy and how it fits into the thesis of the rest of the paper. Mining (a major special interest and strong economic force) is the antithesis of a circular and regenerative economy. The innovations around regenenative economies need to be social, political and cultural much more than technological. The article hints at the cultural element in section 5 but only argues that there will be a more urban focus - not how this focus will be regenerative. Cities are highly intensive users of resources, even if they become more efficient smart cities. Can cities ever be regenerative, or will they continue to rely on their regions for some of the regeneration (and if so are they willing to pay for it).

Overall, it is not clear how the technological and innovative changes to the city fabric the author discusses in detail encourage the regenerative shift presented. They are sort of implied by the circular economy and biophilic cities. In fact, a circular economy and zero-carbon might not be enough to get us to regenerative. As the author notes at one point, regenerative means going beyond zero carbon, but the Paris agreement on which this transition is to be based really only takes us to (net)-zero (if we are lucky!).

The transition presented is based around following the SDGs and Paris Agreement. This is a big assumption, especially the second one (see the US!). I think it should be presented as a hypothesis or assumption rather than a 'will'. The pandemic could lead to countries isolating themselves more and the weakening of international structures putting these very agreements under threat. Also, international agreements can simply lend themselves to maintaining a status quo, due to the way they are negotiated, rather than encouraging transition.

Similarly, the importance of equity is mentioned at the start and the end. But there's little discussion of how the innovations might help (or not) with equity. I'd argue the equity is one of the major challenges and barriers we face, and all the technological innovations we have actually risk increasing inequity. This is a common critique of Smart Cities - who has access, who has control? I'm not sure that a circular or even regenerative economy necessarily has 'equity' built in, although possibly it is included in some definitions. A circular economy could still be very unequal - I get to recycle my things, but badly paid immigrants have to take them apart and separate the components to be reused!

I don't think the author necessarily needs to address or discuss in detail all the challenges (there is a word limit after all!). I also don't think they can just be lumped together as barriers to what is an 'expected' transition.
The author could introduce a more critical lens to the technologies and innovations presented and note the challenges they bring with them. Noting, for example, the critiques of Smart Cities, the difficulties of transitioning to renewable energy in a world dominated by fossil fuels, the equity issues around mobility etc. There is some of this there, but I think it needs to go further.

An alternative is to adjust the 'framing' of the article, as discussed above. If the article is framed as discussing opportunities, for example, then the author could state this upfront, acknowledge its 'positive' view and the mention of challenges can come towards the end - expanded a little. Similarly, it could be framed as a vision but again this needs to be made clear. However, if the article is really about the 'expected' transitions of the title, then the barriers and challenges (and risks) and how they are going to be overcome in this expected transition need to be made more clear throughout.

Summary and suggestions

Overall these separate critiques lead me to suggest that:
1. The paper is less about 'expected' transitions than possible, positive transitions and this needs to be clear
2. The argument for the links between renewable energy, smart cities and electromobility and the move towards globalised-localism are strong. However, the links between the innovations discussed, the changes in urban fabric called for and the move towards regenerative economy are not clear enough and need improving for publication.

To me, the author is coming very much from an ecological mondernist perspective. Perhaps most notable when discussing the 'decoupling' of carbon emissions from economic growth at the start of section 3 (as an aside decoupling has been critiqued and can not be taken as possible let alone likely). I can see how this approach could lead us towards the gobalised-localism vision presented here, with the renewable energy, smart city technology and changes in mobility helping to change the urban fabric, which in turn encourages innovation in these areas. However, a transition to a regenerative economy is, I suggest, much more towards a deep ecology perspective. I see too big a step from some of the opportunities technology provides around renewable energy (including hydrogen), electromobility and smart cities to a regenerative economy. The circular economy, biophilic urbanism and permaculture are much more aligned to these regenerative ideas, but these receive much less attention, especially in the discussion of the urban fabric. The links are less clear here and the 'expectation' of such a transition seems hopeful at best.

It could be that the article is trying to do too much. It could focus just on the first elements and how they (could) move us towards the globalised-localism. This is especially relevant to the pandemic, which has highlighted both out global connections and our vulnerable reliance on long supply chains and centralised systems. This would be a good article for publication.

However, I would agree that we need to move to a regenerative economy. I wonder if we might need a seventh wave for that (sadly, because the collapse before it could be terrible). To include it the sixth wave needs a greater transition than is implied and discussed in this article. I would suggest, for example, that it is likely incompatible with any form of economic system (e.g. capitalism) we have seen in the modern world thus far. This links to the challenges, which are much greater than for these regenerative transitions than the author discusses. The author could frame the article the following way: there are strong possibilities of a transition to globalised-localism and technology and innovation can be applied to the design of the urban fabric to help this transition. However, we need to go further towards regenerative economies, and while there are hints from biophilic design, permaculture and a circular economy we need a lot more to get there (or there are a lot more barriers).

Minor comments
Table 1 is titled "The six waves of innovation" but in numbered 1 to 5. This is really presenation - the table is numbered by collapses rather than waves.

You could include 3.5 Hydrogen-based industry into renewable energy (on the assumption the hydrogen is produced using renewable energy). One way of looking at hydrogen is simply a storage and transport mechanism for renewable energy - which you imply already.

In discussing decoupling, you refer to a work on theology of sustainability. I think there are other works more specifically looking at decoupling that could be referenced.

I think there is scope to refer back to the impacts of the pandemic a little more throughout. The argument is that the pandemic and related crash will lead to this wave of innovation. I think there is a strong link between what the pandemic has shown us and the globalised-localism.

Reviewer 2 Report

Comments to author:

Paper can benefit from creating and adding graphs such as what will happen when we go back to business as usual vs with zero carbon strategies for economy and carbon emissions

Paper can also benefit from adding suggestions on how to introduce such strategies to the administrations and ways in which city can adopt in terms of using the available resources

Paper can also benefit from adding numbers such as jobs that will be created, economic growth that can happen and may be some measure of standard of living

Paper should include by adding the technical adaptions required, time for adaptation and what can a city do while it is adapting. Suddenly new concepts cannot be implemented or jump-started forgetting what is already there.

Paper suggests that after COVID 19 cities have a clean slate to start adapting zero carbon and zero poverty strategies. Paper should mention why and how it is that. It is more likely people will go back to business as usual after the pandemic is over.

Huge concepts. Mostly all disconnected. the author is off course very learned and knows many concepts but unfortunately has not knit these concepts together. Just have put all these new concepts bundled up in this paper.

Paper doesn't have a research design and has a lot of jargon. Concepts are disconnected. The paper needs to be re-written with a fresh eye. 

Reviewer 3 Report

The author addresses a determinant theme with profound implications for the environmental crisis that the world is going through: the change in global development paradigms (expressed in a sixth wave of global innovation).

The urgency for the implementation of SGDs, the need to define their goals and the ways to achieve them, as well as equity and quality of life are, admittedly, key themes that surround us and that currently influence the way of thinking about the future global.

None of these issues is new, nor does the article set out to reflect on a new global paradigm. On the contrary, the article approaches old themes very intelligently and in a very assertive way and relates them to the sudden acceleration of these dominant paradigms caused by the pandemic crisis of 2020. The overall opinion is that the text is very well written and with a remarkable narrative.

Nevertheless there are small details needing furhter explanation.

Although I fully agree with the author, I believe that this article could reflect on the risk of the delay in the “sixth wave”, that is, it is not clear that the acceleration of a change caused by the current world crisis embraces the wave of green economics. Is it possible that the disruption of the current system caused by the pandemic could delay innovation instead of accelerating the global innovation process?

On page 5 the author mentions “Old coal and gas power in the Covid downturn are being avoided as new solar and battery systems have marginal costs of zero thus the reductions in demand must be made by fossil fuel-based power”. The phrase is paramount to understand the post-covid energy changes but I would like to see a source for this information.

Another important concept that could be more developed has to do with the circular economy. There are issues that promise to be contradictory and it is important to reflect: If it is understandable that smart cities will be fundamental in the process of change, their relationship with the role of the productive system is less clear. In a circular economy logic, should local products “feed” cities? How will food and goods be produced? Within or in connection (but separate) from compact cities? What implications does this system have on spatial distribution of urban areas?

Are we heading towards polynuclear cities? Are we moving towards the emergence of medium-sized cities that are linked together and are separated by production areas?

The approach to the circular economy in the article is very important but somehow disproportionate to the other themes. Perhaps it could have gone a little further as the concept is fundamental to the point 4 of the article, where the author focuses exactly on the issues raised here.

Reviewer 4 Report

Dear author, 

Your paper entitled COVID, CITIES and CLIMATE:Historical Precedents and Expected Transitions for the New Economy, brings new challenges but also opportunities in the era post COVID-19, and could be the basis for new new targets in the sustainable development goals, whereas government, scientific and communities are involved. The work highlights the importance of enabling long term trends for our cities in terms of climate agenda, however, there is room for improvement before is acceptable for the publication in the Journal of Urban Science.

My concerns are as follows:

  • What are the roles of different stakeholders play in this new post-COVID-19 scenarios? Could you describe, differentiate these roles?
  • I would like to summarize the main challenges and possible solutions you proposed in one table, it could help the lector to easily understand all your work 
  • I understand that this paper is not about the economic consequences of post-COVID-19, however, I would like to shortly discuss the economic crisis for countries and the challenges to achieving the new transition, particularly for developing countries 

Best regards

The reviewer
